# Substance Use Problems and Death of Despair in a 32-Year Follow-Up Study of Suicide Attempters Examined at a Medical Emergency Inpatient Unit

**DOI:** 10.3390/ijerph22040575

**Published:** 2025-04-06

**Authors:** Erik Niwhede, Jonas Berge, Agneta Öjehagen, Sara Lindström

**Affiliations:** 1Department of Clinical Sciences, Lund, Psychiatry, Lund University, 22185 Lund, Sweden; erik.niwhede@med.lu.se (E.N.); jonas.berge@med.lu.se (J.B.); agneta.ojehagen@med.lu.se (A.Ö.); 2Addiction Center Malmö, 20502 Malmö, Sweden; 3The Region Skåne Committee on Psychiatry, Habilitation and Technical Aids, 22185 Lund, Sweden

**Keywords:** death of despair, suicide, suicide attempters, substance use disorders, long-term follow-up

## Abstract

Deaths of despair (DoD), encompassing suicide, drug overdose, and alcohol-related mortality, are often linked to social and psychological distress. This study examined the long-term risk of DoD among individuals previously hospitalized for suicide attempts, with a focus on substance use problems and psychiatric disorders as potential risk factors. A prospective cohort of 1044 individuals admitted to inpatient care following a suicide attempt between 1987 and 1998 was followed for up to 32 years using national registers. Cox proportional hazards regression models were used to assess associations between psychiatric diagnoses and mortality outcomes. The results showed that substance use problems, compared to no such problems, were associated with an increased risk of all-cause mortality but not DoD. In secondary analyses, dysthymia was linked to an increased risk of suicide but not non-suicidal DoD, suggesting distinct underlying mechanisms. Notably, aside from gender, no control variables were significantly associated with non-suicidal DoD, indicating that other factors may play a more prominent role in this high-risk population. These findings challenge the traditional DoD framework and highlight the need for tailored prevention efforts that consider the distinct risk profiles of suicide and non-suicidal DoD. Future research should incorporate socioeconomic and demographic factors to enhance understanding and prevention strategies.

## 1. Introduction

The risk of suicide death among individuals with a history of suicide attempts represents a critical public health challenge. Long-term studies have shown that between 2% and 13% of individuals who attempt suicide eventually die by suicide [1,2,3]. A recent systematic review and meta-analysis by Pemau et al. [4] identified several key risk factors for suicide reattempts, including prior suicide attempts, active suicidal ideation, trauma, alcohol misuse, and drug misuse. Similarly, a review by Mendez-Bustos et al. [5] emphasized that recurrent suicide attempters are more likely to experience psychiatric disorders, stressful life events, and socioeconomic disadvantages, such as unemployment and being unmarried. Furthermore, de la Torre-Luque et al. [6] highlighted the multidimensional nature of risk, with psychosocial vulnerabilities like trauma and social isolation compounding the likelihood of repeated attempts.

Co-occurring Substance Use Disorders (SUDs) and psychiatric disorders are highly prevalent, with over 50% of psychiatric inpatients having a co-occurring SUD, and more than half of individuals with SUD meeting criteria for another psychiatric disorder, such as anxiety, mood, personality, and schizophrenia/psychotic disorders [7]. These dual diagnoses often exacerbate symptoms, increase the risk of adverse outcomes, and complicate treatment approaches [8]. A systematic review of individuals seeking treatment for SUD in Australia found that comorbid conditions were linked to more severe substance use, poorer social and occupational functioning, and increased risk of homelessness [9].

Individuals with SUDs face significantly higher mortality risks compared to the general population. A longitudinal study [10] reported a mortality risk nearly four times higher than the general population. Patients with alcohol use disorder (AUD) alone primarily died from somatic diseases, while those with poly-substance use disorders were more likely to die from overdoses. Older age and long-term SUD were key risk factors for mortality. SUDs are also associated with increased risks of suicide. A meta-analysis of over 870,000 participants found that individuals with SUDs had an elevated risk of suicide death (OR 1.49) [11]. Opioid use was particularly associated with higher risks.

The bidirectional relationship between SUDs and suicidality further complicates the clinical picture. A systematic review by Rioux et al. [12] noted that individuals with SUDs are not only at an increased risk of suicidal thoughts and behaviors but that those experiencing suicidal ideation may also turn to substances as a coping mechanism. This reciprocal influence makes it challenging to determine the precise intent behind deaths in this population. Substance use also plays a critical role in the classification of suicides and deaths of undetermined intent. A Swedish register-based study [13] found that deaths classified as undetermined intent were more likely to involve individuals with a history of hospitalization for substance abuse compared to those classified as suicides. Poisoning was a common method in undetermined intent deaths, further complicating intent determination. Similarly, a Canadian study [14] observed that while suicide rates involving poisoning declined by 1.2% annually from 2000 to 2011, unintentional and undetermined poisoning deaths increased significantly during the same period, with drug-related poisonings, particularly involving opioids, dominating these categories.

These findings underscore the interconnected nature of suicidality, substance misuse, and social determinants of health, creating a complex landscape of risk that extends beyond traditional categorizations of suicide or accidental deaths.

The concept of death of despair (DoD), introduced by Case and Deaton [15] provides a comprehensive framework for understanding these overlapping phenomena. Defined as suicides, drug-related deaths, and alcohol-related deaths, this category captures the shared underlying causes driven by psychological distress and social despair, including financial hardship, social isolation, and hopelessness. By framing these outcomes within the broader concept of deaths of despair, we can better capture the shared etiological pathways underlying suicidality, substance misuse, and social determinants of health. The concept of Deaths of Despair (DoD) was originally introduced to describe excess mortality among middle-aged white men in the United States, primarily driven by suicide, drug overdose, and alcohol-related causes [15]. However, the applicability of DoD to broader populations, particularly individuals with psychiatric conditions, remains underexplored. While the DoD framework captures deaths linked to socioeconomic distress and behavioral health crises, it does not fully account for other important causes of premature mortality among individuals with psychiatric disorders, such as cardiovascular disease, metabolic disorders, and homicide.

Despite extensive research on suicide risk and substance use disorders (SUDs), several gaps remain. While prior studies have identified key risk factors for suicide reattempts and mortality among individuals with SUDs, the interplay between psychiatric disorders, SUDs, and social determinants in predicting long-term outcomes such as deaths of despair remains poorly understood. Additionally, the classification of deaths, particularly poisoning-related deaths of undetermined intent, complicates efforts to assess the true burden of suicide and related outcomes accurately. Limited evidence exists on how hospitalization for suicide attempts interacts with comorbid psychiatric and substance use disorders to influence subsequent mortality risks. Addressing these gaps is critical for developing more precise prevention and intervention strategies.

The primary aim of this study is to assess the impact of substance use problems on deaths of despair (DoD), suicide, and all-cause mortality among individuals with a history of hospitalization for suicide attempts. By focusing on substance use problems, the study aims to provide a deeper understanding of their role in shaping mortality outcomes in this high-risk population. The secondary aims are to:Evaluate the overall risk of DoD, suicide, and all-cause mortality in this population.Investigate whether psychiatric disorders, in addition to substance use disorders, independently contribute to these mortality outcomes.

## 2. Materials and Methods

### 2.1. Study Design and Study Population

This study represents a follow-up of a previously examined population and is based on clinical interview data at the suicide attempt combined with register information from a large cohort (*n* = 1044) of individuals who were hospitalized following a suicide attempt. These individuals were admitted to a medical emergency inpatient unit (MEIU) in southern Sweden, where they underwent comprehensive assessments conducted by a psychiatrist and a social worker. They were subsequently followed up for a period of 21 to 32 years.

The sample was recruited between 1987 and 1998 and accounts for approximately half of all individuals admitted to the MEIU due to a suicide attempt during that period (estimated 1065 out of 2000 cases [16]). Individuals admitted on weekends or holidays were not included in the study because the research team was not on duty during weekends, nor were those who had already been enrolled and later reattempted suicide. On weekdays, all individuals who attempted suicide underwent standardized assessments using specific evaluation measures (for further details, see [17]). As most admissions occurred on weekdays, a subset of individuals admitted during weekends (*n* = 251) was compared to the rest of the cohort. Analyses revealed no statistically significant differences in suicide frequency or overall mortality between the two groups in both the previous follow-up study and the present sample.

Data regarding mortality, including date of death and emigration, were obtained from the Swedish Tax Agency, covering the period until July 2019. Information on causes of death was retrieved from the Swedish Cause of Death Register (CDR), maintained by the National Board of Health and Welfare [18] covering the study period from January 1987 to 31 December 2018. Initially, the study aimed to include all 1052 individuals from the first follow-up [17]; however, eight individuals were lost to follow-up by 2019, and another 6 were removed due to missing data for the death date in the CDR, resulting in a final sample of 1038 participants.

The mean follow-up time was 20 years (range 0–32 years), with a median of 22 years and 10 months, totaling 20,887 person-years. The mean age at the time of the index attempt was 39.8 years (range 15–92 years).

### 2.2. Baseline Investigation

The baseline data were collected at emergency psychiatric consultations that were always requested of patients who have attempted suicide in the medical emergency inpatient unit. An assessment protocol based on a semi-structured interview that was developed by the Suicide Research Centre in the Department of Psychiatry at Lund University was in use from 1987 to 1998 [19]. This protocol was used to improve the procedure for treatment referral with the help of a broad and standardized manner of assessment by a psychiatrist and a social worker, which included sociodemographic data, method of suicide attempt, occurrence of previous attempts, and suicidal intent. Psychiatric disorders were diagnosed by clinical interviewing, but no structured interview was carried out for diagnosis, and in evaluation of symptoms was assessed in accordance with DSM-III-R.

### 2.3. Variables

#### 2.3.1. Baseline Variables

Psychiatric diagnosis, age, gender, and the SAD PERSONS questionnaire were assessed at admission at the medical emergency inpatient unit through clinical interviewing. A suicide attempt is characterized as an act where an individual engages in behavior that is either genuinely or apparently life-threatening, to endanger their life or create the impression of such an intent, without resulting in death. Psychiatric disorders were diagnosed using DSM-III-R [20]. Only the main diagnosis was used for the study due to very high rates of missing data concerning any secondary diagnosis at the acute consultation. Substance use in this study was defined as either a diagnosed substance use disorder (SUD) according to DSM-III-R criteria and/or clinician-rated excessive alcohol or drug use as indicated by the SAD PERSONS scale [21]. The SAD PERSONS scale is a widely used screening tool designed to assess suicide risk by evaluating ten key risk factors commonly associated with suicidal behavior, including Sex, Age, Depression, Previous suicide attempts, Excessive alcohol or drug use, Rational thinking loss, Social support deficits, Organized plan, No spouse, and Sickness.

However, in this study, the SAD PERSONS scale was not employed to assess suicide risk. Instead, it was utilized to capture clinician-assessed indications of alcohol or drug-related problems that may not have been formally diagnosed. Study participants who either met the criteria for SUD using DSM-III-R and/or were classified as having excessive substance use in the SAD PERSONS screening tool were categorized as having substance use problems. This dual approach allowed for the identification of individuals with substance use problems that might have otherwise remained undetected through diagnostic records alone.

#### 2.3.2. Outcome Variables

Death of Despair contains several variables from ICD-10 and ICD-9 that describe the investigated phenomena of death of despair and are, in our judgment, the best translation of what the original creators of the concept of death of despair intended. These include suicide (see definition below), uncertain suicides (see definition below), diseases of the liver (ICD-10: K70) and accidental poisoning by drugs (ICD-10: X40–X45), drug dependence (ICD-9: 3040), Cirrhosis due to alcohol (ICD-9: 5712) and toxic effects of alcohol (ICD-9: 9800–9828).

Suicide was defined based on classifications in the Cause of Death Register, referring to external causes of morbidity and mortality due to intentional self-harm, corresponding to ICD-10 codes X60–X84 and ICD-9 codes beginning with E95. Deaths of undetermined intent, often referred to as uncertain suicides, were classified in the Cause of Death Register under external causes of morbidity and mortality, following ICD-10 codes Y10–Y34 and ICD-9 codes starting with E98The variable Non-suicide death of despair contains all the variables from death of despair, excluding suicide. To aid our analysis, the ICD-9 codes were translated into ICD-10. This was done manually with one ICD-9 code at the time. Lastly, the variables and the translated ICD-10 codes are presented in Table 1.

Natural causes of death, which encompassed diseases of the circulatory system (ICD-10: I00–I99), respiratory system (ICD-10: J00–J99), digestive system (ICD-10: K00–K93), nervous system (ICD-10: G00–G99), endocrine, nutritional, and metabolic diseases (ICD-10: E00–E90), neoplasms (ICD-10: C00–D48), certain infectious and parasitic diseases (ICD-10: A00–B99), congenital malformations and chromosomal abnormalities (ICD-10: Q00–Q99), and other ill-defined conditions (ICD-10: R00–R99). These cause-of-death classifications align with those presented in Table 1 and provide a comprehensive framework for analyzing mortality patterns in this cohort.

### 2.4. Data Analysis

Cox proportional hazards regression models were employed to investigate the associations between baseline characteristics and mortality outcomes. These models were chosen due to their suitability for analyzing time-to-event data, allowing us to estimate hazard ratios (HRs) and their corresponding 95% confidence intervals (CIs) for all-cause mortality, suicide, deaths of despair, and non-suicide deaths of despair. Time-to-event was defined as the period from the date of hospital admission for the index suicide attempt to the occurrence of the event of interest or the end of the follow-up period. Individuals who did not experience the outcome were censored at the date of emigration or the end of the observation period.

Separate Cox models were estimated for each outcome variable, with adjustments made for potential confounders identified based on prior literature and clinical relevance. The covariates included in the adjusted models were sex, psychiatric diagnosis (major depressive disorder, adjustment disorder, dysthymia, psychosis), substance use disorder, and others. Age at suicide attempt was also included in the regression models, though not as an independent variable but through the use of age as a timescale, which provides full adjustment of the effect of age on the outcome measures, though no numerical estimate is provided by the model. The proportional hazards assumption was evaluated using Schoenfeld residuals, and no violations were detected.

Kaplan-Meier survival curves were constructed to visually assess the cumulative incidence of each outcome across diagnostic categories and sex. A significance level of *p* < 0.05 was considered statistically significant for all analyses. All statistical analyses were performed using R (version 4.4.0).

## 3. Results

In total, 1038 individuals were included in the study. Of them, 409 (39.4%) were men, and 629 (60.6%) were women. In the total sample, 270 of the individuals (26%) were assessed as having issues with substance use, 170 (16.4%) with Adjustment disorder, 47 (4.5%) with Dysthymia, 329 (31.6%) with Major depressive disorder, 69 (6.6%) with psychosis, 159 (15.2%) with other psychiatric diagnosis which included depression not otherwise specified, anxiety disorder, other axis I-diagnosis, no diagnosis, axis II-diagnosis (excluding axis I-diagnosis) and cases with missing data for diagnosis. At follow-up, 379 of these individuals were dead; underlying causes of death are displayed in Table 1.

Table 2 displays the distribution of demographic and diagnostic variables in the total sample and their association with all-cause mortality. Men had a significantly higher risk of death compared to women, with nearly double the adjusted hazard. Among diagnostic categories, substance use problems were significantly associated with an increased mortality risk, while other disorders, such as adjustment disorder, dysthymia, and psychosis, did not show statistically significant associations in the adjusted models. The survival curves of individuals with and without substance use problems are shown in Figure 1.

Table 3 presents the analysis of risk factors for death of despair. Men had a significantly higher risk of death of despair compared to women, with more than twice the adjusted hazard. Among the diagnostic categories, individuals with dysthymia showed a significant association with death of despair in the adjusted model. Substance use problems, adjustment disorder, and psychosis were not significantly associated with the death of despair after adjustment.

Risk factors for suicide were also investigated, as can be seen in Table 4. Men had a significantly higher risk of suicide compared to women in the adjusted model. Among the diagnostic categories, adjustment disorder, dysthymia, and psychosis were significantly associated with an increased risk of suicide after adjustment for confounders, with dysthymia showing the highest hazard. Substance use problems and major depressive disorder were not significantly associated with suicide risk in the adjusted model.

Risk factors for deaths of despair, excluding suicide, over the 32-year follow-up period were investigated, as shown in Table 5. The findings indicate that men had a significantly higher risk compared to women, with nearly four times the adjusted hazard. Substance use problems were a significant risk factor for non-suicidal deaths of despair, with a strong association even after adjusting for confounders. In contrast, other diagnostic categories, including adjustment disorder, dysthymia, and major depressive disorder, did not show statistically significant associations. No cases of non-suicidal deaths of despair were observed among individuals with psychosis, resulting in undefined hazard ratios.

## 4. Discussion

This study examined the long-term risk of deaths of despair, suicide, and all-cause mortality among individuals previously hospitalized for a suicide attempt and explored psychiatric and substance use problems as potential risk factors. The findings indicate that substance use problems were associated with an increased risk of all-cause mortality and non-suicidal deaths of despair but were not significantly linked to an elevated risk of suicide. Men had a significantly higher risk of both suicide and deaths of despair compared to women. None of the psychiatric diagnoses or substance use problems was a risk factor for deaths of despair, whereas major depressive disorder appeared to be associated with a lower risk. These results suggest that different psychiatric diagnoses contribute differently to various mortality outcomes, highlighting the importance of targeted prevention efforts for this vulnerable population.

A key finding of this study is the significant association between substance use problems and increased mortality risk. Individuals with substance use problems had a markedly higher risk of all-cause mortality, consistent with previous research highlighting the detrimental effects of long-term substance misuse on physical health and overall survival [9]. While substance use problems were not significantly associated with an increased risk of suicide, they were linked to an elevated risk of non-suicidal deaths of despair, such as drug overdoses and alcohol-related diseases [10]. This finding highlights the complex relationship between substance use problems and mortality risk, indicating that while substance use problems may not coincide with higher suicide risk, it is associated with increased long-term health-related mortality [11]. Several explanations may account for this discrepancy. First, our sample exclusively comprised individuals hospitalized after a suicide attempt, indicating an already heightened suicide risk regardless of substance use status. The homogeneity of this high-risk group could thus reduce the observed impact of substance use problems as an isolated factor. Second, given the long follow-up period, it is possible that changes in substance use or access to treatment services over time attenuated the association between initial SUD diagnosis and subsequent suicide. Third, methodological factors such as diagnostic practices at baseline or underdiagnosis of less severe substance use issues at the time of hospital admission could also influence our findings.

Dysthymia, characterized by persistent depressive symptoms, was found to significantly increase the risk of suicide but not death from despair, distinguishing it from previous studies that have linked chronic depressive states to substance-related mortality [15]. These findings highlight the need for improved clinical screening and early intervention strategies to identify and manage at-risk individuals [22].

Consistent with prior research, our study found that men had a significantly higher risk of suicide compared to women, which aligns with previous findings showing that men who attempt suicide are at greater risk of subsequent suicide death [17]. However, the broader category of deaths of despair, which includes drug- and alcohol-related deaths, may be influenced by additional factors that warrant further investigation. Gender differences in mortality outcomes may reflect variations in coping mechanisms, social support networks, and healthcare-seeking behaviors. Case and Deaton have suggested that men are less likely to seek help for mental health issues and may rely on maladaptive coping strategies, such as substance use, which can contribute to higher rates of deaths of despair [15]. This underscores the importance of addressing gender-specific barriers to care and designing targeted interventions that consider the unique vulnerabilities of men and women in relation to substance use problems and mental health. Moreover, research on social integration and psychological distress has shown that men with weaker social networks and limited access to mental health services may face an increased risk of death or despair [23].

### 4.1. Implications for the Concept of Deaths of Despair

Our findings indicate that, with the exception of gender, none of the included control variables were significantly associated with the death of despair (DoD). This could partly be explained by the fact that these variables were measured many years before the deaths occurred, meaning that changes in individuals’ life circumstances and health status over the follow-up period were not captured. However, it is important to note that these variables demonstrated clear associations when DoD was analyzed separately for suicide and non-suicidal DoD. This suggests that while they may not predict DoD as a whole, they still play a significant role in distinguishing between these different outcomes. Further, the results may suggest that in a highly vulnerable patient group, such as individuals hospitalized after a suicide attempt, other factors may better explain DoD than traditional demographic and clinical variables.

The concept of “deaths of despair” (DoD), encompassing suicide and deaths related to drugs and alcohol, captures diverse expressions of underlying existential and social despair. DoD emphasizes that regardless of the specific cause of death, these deaths stem from a fundamental sense of hopelessness and an unsustainable life situation. Interestingly, our findings both support and challenge aspects of this theoretical perspective. While we found similar overall risks of DoD between individuals with and without substance use disorders—supporting the notion of a common underlying psychosocial distress—we also observed important distinctions when examining specific outcomes separately (suicide versus non-suicidal DoD). Different psychiatric and demographic variables appeared to uniquely influence suicide and non-suicidal DoD, suggesting that these outcomes might reflect distinct psychiatric and psychosocial phenomena rather than a unified concept in this clinical population, where suicidal behavior and substance-related mortality may have different underlying drivers and risk profiles.

However, future studies would benefit from incorporating detailed socioeconomic and demographic variables, such as education level, as these factors may have a stronger connection to DoD in other populations or broader community settings. Social factors such as financial hardship, loneliness, and lack of social support have been shown in previous research to play a crucial role in these outcomes. They could provide further insights into the interplay between psychiatric and social risk factors.

### 4.2. Strengths and Limitations

This study has several strengths that contribute to its relevance and potential value. One notable strength is the extended follow-up period of up to 32 years, which offers a rare opportunity to explore long-term mortality outcomes in a population with a history of suicide attempts. The large sample size further strengthens the study by allowing for a more comprehensive examination of patterns over time. Moreover, the use of national registry data provides a reliable and objective source of information on mortality, psychiatric diagnoses, and substance use, which enhances the robustness of our findings. By examining both suicide and non-suicidal deaths of despair, the study contributes to a broader understanding of mortality risks beyond traditional suicide research.

A key challenge in research on deaths related to substance use and suicide is accurately determining the intent behind the death, particularly for overdoses classified as having undetermined intent. In our study, deaths categorized as having undetermined intent (ICD-10 codes Y10–Y34) were not included in the suicide category, potentially leading to an underestimation of actual suicide incidence. This decision highlights broader methodological challenges, as distinguishing intentional self-harm from accidental overdoses often depends on subjective clinical assessments and varying legal definitions.

Additionally, the study’s approach of identifying substance use problems through both clinical diagnoses and clinician-rated indicators from the SAD PERSONS scale offers a more nuanced understanding of substance-related issues. This approach, while not without its limitations, aims to capture individuals who may not have received a formal diagnosis but who still exhibit significant substance use concerns.

Despite these strengths, several limitations should be acknowledged. First, psychiatric assessments were conducted at the time of hospitalization, and the lack of follow-up assessments limits our ability to account for changes in mental health status and treatment effects over time. This is an important consideration, as individuals’ conditions and circumstances may have evolved in ways that were not captured in our data. Further, although a standardized psychiatric assessment was conducted by a psychiatrist at the MEIU, the emergency setting in which it took place may have posed challenges in obtaining a comprehensive psychiatric history. Furthermore, since only primary diagnoses were included in the analysis, it is possible that participants met the criteria for additional diagnoses not considered in this study. Therefore, these factors should be taken into account when interpreting the results.

Second, while registry data provides a comprehensive overview, it may not capture the full complexity of individuals’ experiences. Factors such as undiagnosed psychiatric conditions or unreported substance use could lead to an underestimation of their true prevalence and impact.

Furthermore, it is important to acknowledge that the study population consists of individuals who were hospitalized following a suicide attempt, which may limit the generalizability of the findings to individuals with less severe suicidal behavior who did not require hospitalization. Socioeconomic factors, such as employment status and social support, were not directly assessed, even though these factors are known to influence mental health outcomes and mortality [22]. Future research could benefit from a more detailed consideration of these aspects. Substance use problems were common among the cohort; however, we did not have access to data specifying the type of substances used, duration of use, or previous treatment for substance dependence. Future research should examine these factors in more depth to assess their impact on mortality risk.

Another limitation of this study is the absence of a control group consisting of individuals treated in the emergency department without a suicide attempt. While a comparison group would have provided important insights into differences in mortality causes, such an analysis was not feasible within the available dataset. Future studies should consider including a control group to better isolate the specific impact of prior suicide attempts on mortality risk.

A limitation of the statistical methods is that the low number of events for some of the outcome variables required us to limit the number of variables included in the regression models. We, therefore, chose to omit some potentially important variables such as previous suicide attempts, method of suicide attempt, marital status, and employment. Furthermore, in the analysis of non-suicide deaths of despair, the number of events per independent variable was very low, which makes statistical inference somewhat more difficult.

Finally, while efforts were made to adjust for key confounding variables, residual confounding cannot be ruled out. Variables such as genetic predisposition, early-life adversity, and access to healthcare services were not accounted for and may have influenced the observed associations. We recognize the complexity of the interplay between these factors and encourage future studies to build upon our findings with more comprehensive models.

### 4.3. Future Directions

The findings of this study highlight several areas that warrant further investigation. First, future research should aim to explore the longitudinal trajectories of individuals with a history of suicide attempts, with a particular focus on how changes in psychiatric and substance use problems over time influence mortality outcomes. Understanding the dynamic nature of these risk factors could provide valuable insights into targeted interventions at different stages of an individual’s life.

Second, a more nuanced examination of the social determinants of health, such as socioeconomic status, employment status, social support networks, and access to healthcare, is needed to better understand their role in deaths of despair. Given the established link between social adversity and mental health outcomes, incorporating these factors into future studies could help identify key intervention points that may mitigate long-term mortality risks. Future research should also explore substance use in greater detail, including specific substances, consumption patterns, and treatment history, to clarify their role in Deaths of Despair (DoD)-related mortality.

Additionally, future studies should consider integrating qualitative approaches to complement quantitative findings. Exploring the lived experiences of individuals with psychiatric and substance use disorders may provide a deeper understanding of the barriers to seeking care and maintaining recovery, which could inform the development of person-centered prevention strategies.

Moreover, refining the classification and reporting of causes of death is essential to ensure more accurate mortality estimates. The differentiation between suicide, accidental deaths, and undetermined intent remains a challenge, and future efforts should aim to enhance the reliability of death classifications, potentially through multi-source data integration and improved forensic assessments.

An analysis of the risk factors associated with natural deaths in individuals with a history of SUD, excluding DoD-related deaths, would be valuable, but such an investigation falls outside the primary scope of this study. However, future research should further explore the broader context of natural deaths among individuals with a history of SUD to provide a more comprehensive understanding of their mortality risk.

Although this study did not explicitly examine factors contributing to the onset or persistence of substance use disorders, future research could benefit from a more comprehensive analysis of such factors. Including detailed information on substance use patterns (e.g., severity, type of substance) and relevant social determinants (e.g., education, marital status, housing, income, social networks) would provide valuable insights into their potential role as confounders and deepen our understanding of the complex interactions that shape mortality risks in this vulnerable population.

Finally, intervention-focused research is crucial to evaluate the effectiveness of existing prevention strategies and to develop new approaches tailored to high-risk populations. Randomized controlled trials and implementation studies focusing on integrated psychiatric and substance misuse interventions could offer practical insights into reducing the incidence of deaths of despair.

## 5. Conclusions

This study provides valuable insights into the long-term mortality risks among individuals previously hospitalized for a suicide attempt, emphasizing the distinct pathways leading to suicide, deaths of despair (DoD), non-suicidal death of despair, and all-cause mortality. Our findings highlight the complexity of these outcomes, revealing that different psychiatric disorders contribute to specific mortality risks. While substance use problems were strongly associated with an increased risk of non-suicidal DoD and all-cause mortality, they were not linked to suicide risk. Conversely, dysthymia was a significant predictor of suicide but did not increase the risk of non-suicidal DoD, suggesting that suicide and other despair-related deaths may have distinct underlying mechanisms.

Notably, with the exception of gender, none of the included control variables were significantly associated with non-suicidal DoD, indicating that substance use problems, as well as other unmeasured factors—such as long-term social and economic circumstances—may better explain these outcomes. These findings challenge the DoD framework, which views suicide and substance-related mortality as part of the same phenomenon and underscores the need for a more nuanced approach that recognizes their differences.

Our results also underscore the importance of addressing substance use problems in reducing overall mortality, given their strong association with all-cause death. This highlights the need for integrated healthcare approaches that combine psychiatric and addiction treatments to reduce long-term mortality risks in this high-risk population.

## Figures and Tables

**Figure 1 ijerph-22-00575-f001:**
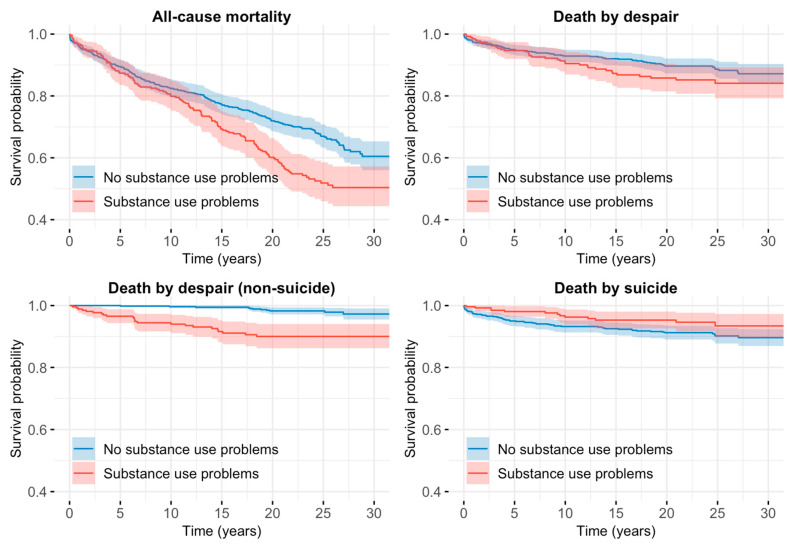
Displays survival curves of all-cause mortality, death of despair, death of despair (non-suicide), and death by suicide.

**Table 1 ijerph-22-00575-t001:** Underlying causes of death (*n* = 379).

ICD 10	Diagnostic Category	*n*	%
	Natural causes of death (*n* = 244, 64.4%)		
A00–B99	Certain infectious and parasitic diseases	13	3.4
C00–D48	Neoplasms	58	15.3
E00–E90	Endocrine, nutritional and metabolic diseases	6	1.6
F00–F99	Mental and behavioural disorders	9	2.4
G00–G99	Diseases of the nervous system	8	2.1
I00–I99	Diseases of the circulatory system	101	26.6
J00–J99	Diseases of the respiratory system	32	8.4
K00–K93	Diseases of the digestive system	7	1.8
M00–M99	Diseases of the musculoskeletal system and connective tissue	2	0.5
N00–N99	Diseases of the genitourinary system	2	0.5
R00–R99	Symptoms, signs and abnormal clinical and laboratory findings not elsewhere classified	13	3.4
Q00–Q99	Congenital malformations, deformations and chromosomal abnormalities	2	0.5
	Unnatural causes of death (*n* = 135, 35.6%)		
Death of despair			
X60–X84	Intentional self-harm (Suicide)	73	19.3
	Non-suicidal death of despair	35	9.2
Other			
X85–Y09	Assault	1	0.3
V01–X59	Accidents (X40–X49 excluded)	8	0.5
Y16–Y34	Event of undetermined intent	5	2.1
Y35–Y99	Other external causes	4	1.1
Total		379	100

**Table 2 ijerph-22-00575-t002:** Risk factors for all-cause death after 0–32 years—(Cox regression).

Variables	Death *n* (%)	uHR (95% CI)	aHR (95% CI)	*p*
Sex				
Men	188 (45.7)	1.92 (1.56–2.37)	1.99 (1.60–2.48)	0
Women (reference)	191 (30.2)	1	1	-
Diagnostic categories				
Substance use problems	126 (46.5)	1.50 (1.18–1.91)	1.42 (1.06–1.89)	0.018
Adjustment disorder	83 (48.5)	1.19 (0.91–1.57)	1.21 (0.82–1.80)	0.335
Dysthymia	26 (55.3)	1.15 (0.74–1.76)	1.40 (0.81–2.41)	0.227
Major depressive disorder	77 (23.3)	0.57 (0.44–0.75)	0.71 (0.47–1.07)	0.098
Psychosis	32 (46.4)	1.36 (0.96–1.93)	1.51 (0.95–2.39)	0.083
Other	59 (37.1)	1.01 (0.77–1.31)	1.02 (0.69–1.50)	0.915

**Table 3 ijerph-22-00575-t003:** Risk factors for death of despair after 0–32 years—(Cox regression).

Variables	Death of Despair *n* (%)	uHR (95% CI)	aHR (95% CI)	*p*
Sex				
Men	58 (14.1)	2.07 (1.41–3.04)	2.23 (1.50–3.32)	0
Women (reference)	50 (7.9)	1	1	-
Diagnostic categories				
Substance use problems	36 (13.3)	1.36 (0.89–2.06)	1.38 (0.84–2.26)	0.207
Adjustment disorder	27 (15.8)	1.75 (1.10–2.80)	1.73 (0.88–3.39)	0.113
Dysthymia	9 (19.1)	1.91 (0.92–3.95)	2.39 (1.00–5.72)	0.051
Major depressive disorder	20 (6.1)	0.37 (0.23–0.61)	0.54 (0.26–1.15)	0.109
Psychosis	12 (17.4)	1.65 (0.90–3.02)	1.69 (0.77–3.71)	0.194
Other	15 (9.4)	0.88 (0.50–1.53)	0.95 (0.45–2.01)	0.892

**Table 4 ijerph-22-00575-t004:** Risk factors for suicide after 0–32 years—(Cox regression).

**Variables**	**Suicide *n* (%)**	**uHR (95% CI)**	**aHR (95% CI)**	** *p* **
Sex				
Men	34 (8.3)	1.58 (0.99–2.52)	1.78 (1.10–2.87)	0.018
Women (reference)	39 (6.2)	1	1	-
Diagnostic categories				
Substance use problems	13 (4.8)	0.59 (0.32–1.09)	0.74 (0.37–1.47)	0.389
Adjustment disorder	23 (13.5)	2.38 (1.41–4.03)	3.21 (1.17–8.81)	0.023
Dysthymia	8 (17)	2.56 (1.16–5.65)	4.54 (1.43–14.40)	0.010
Major depressive disorder	11 (3.3)	0.28 (0.15–0.53)	0.64 (0.21–1.94)	0.428
Psychosis	12 (17.4)	(2.57 (1.37–4.81)	3.57 (1.24–10.28	0.018
Other	11 (9.9)	0.93 (0.49–1.78	1.58 (0.53–4.7)	0.411

**Table 5 ijerph-22-00575-t005:** Risk factors for death of despair (non-suicide) after 0–32 years— (Cox regression).

**Variables**	**Death of Despair (Non-Suicide) *n* (%)**	**uHR (95% CI)**	**aHR (95% CI)**	** *p* **
Sex				
Men	24 (5.8)	3.89 (1.86–7.67)	3.92 (1.67–9.20)	0.002
Women (reference)	11 (1.7)	1	1	-
Diagnostic categories				
Substance use problems	23 (8.5)	5.35 (2.53–11.31)	4.73 (1.77–12.59)	0.002
Adjustment disorder	4 (2.3)	0.70 (0.24–2.03)	0.78 (0.23–2.68)	0.692
Dysthymia	1 (2.1)	0.63 (0.09–4.40)	0.88 (0.11–7.19)	0.908
Major depressive disorder	9 (2.7)	0.61 (0.28–1.32)	0.82 (0.29–2.38)	0.720
Psychosis	0 (0)	-	-	-
Other	4 (2.5)	0.76 (0.26–2.19)	0.67 (0.20–2.26)	0.496

## Data Availability

Data will be made available at reasonable request.

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
