# Peer review of "Substance Use Problems and Death of Despair in a 32-Year Follow-Up Study of Suicide Attempters Examined at a Medical Emergency Inpatient Unit"

_ijerph, 2025, doi:10.3390/ijerph22040575_

Round 1

Reviewer 1 Report

Comments and Suggestions for Authors

Line 100 "on clinical" instead of "onclinical"

Limitations: Regarding "substance use problems", do you have specific data that could be analyzed in more detail (e.g. types of substances, period of consumption, addiction, presence/absence of detoxification therapies/procedures)? If not, please include in this section.

Future directions: See comment above and include the need for further examination of drug use variables. 

Author Response

Response to Reviewer

We sincerely appreciate the constructive feedback provided by the reviewer. Below, we address each comment point by point and indicate where changes have been made in the manuscript.

  1. Line 100: "on clinical" instead of "onclinical"

Reviewer’s comment 1: Line 100: "on clinical" instead of "onclinical."

Response 1:
Thank you for identifying this error. We have corrected the typo in line 108.

Manuscript change:
Revised "onclinical" to "on clinical" in line 108.

Comment 2: Do you have specific data that could be further analyzed (e.g., types of substances, duration of use, severity, prior detoxification treatment)? If not, please clarify this in the limitations section.

Response 2:
We agree that a more detailed analysis of substance use would be valuable. However, the current dataset does not include information on specific substances, duration of use, or detoxification treatment. We have now explicitly stated this in the limitations section.

Manuscript change:
Added clarification in the limitations section about the lack of detailed data on substance use patterns and treatment history, lines 366-369.

Comment 3: Future directions should consider an analysis of substance use variables, including specific types of substances, patterns of use, and treatment history.

Response 3:
We have now included a discussion in the "Future Directions" section emphasizing the importance of future studies investigating the role of substance use in DoD, including specific substances, usage patterns, and treatment history.

Manuscript change:
Added a statement in "Future Directions" highlighting the need for more detailed analysis of substance-related variables in future research, lines 401-403.

Reviewer 2 Report

Comments and Suggestions for Authors

Dear Authors,

The article is interesting mainly because of the size of the sample.

I recommend adding to line 39 of page 1 the explanation of the acronym SUD (substance use disorders) which then appears on line 79 of page 2

The paper states that there were no statistically significant differences between the group of individuals who attempted suicide during the weekend and the group who attempted suicide during the other days of the week. It is therefore unclear why cases of suicide attempt that occurred during the weekend were not counted in the study.

Author Response

Response to Reviewer

We sincerely appreciate the constructive feedback provided by the reviewer. Below, we address each comment point by point and indicate where changes have been made in the manuscript.

Comment 1: The abbreviation SUD should be defined upon first use in the manuscript.

Response 1:
Thank you for pointing this out. We have now defined "Substance Use Disorder (SUD)" upon its first occurrence in line 39 of the introduction.

Manuscript change: Added the full term "Substance Use Disorder (SUD)" in line 39.

Comment 2: The manuscript states that there were no significant differences between patients who attempted suicide on weekends versus weekdays. Why, then, were weekend cases not included in the study?

Response 2: Weekend cases were excluded because the standardized assessment process used in the study was only conducted on weekdays, due to personnel reasons- the research team performing the assessments were not on duty on weekends.

Manuscript change: Added clarification in the methods section explaining why weekend cases were excluded and included data comparing weekday and weekend cases, lines 116-117.

Reviewer 3 Report

Comments and Suggestions for Authors

This manuscript presents a compelling examination of the characteristics of individuals who attempted suicide and subsequently died under unnatural circumstances.

The introduction is relevant, though it employs the concept of "Death of Despair" (DoD), which has notable limitations. Originally proposed to explain excessive mortality among middle-aged white men, this concept does not adequately account for other types of early deaths linked to psychiatric disorders, such as homicides, cardiovascular conditions, and metabolic diseases.

The authors should provide explanations for any abbreviations used in the introduction, especially for SUD (Substance Use Disorder).

The methodology is clearly articulated; however, the absence of a control group comprising individuals admitted to the emergency department (ED) without suicide attempts restricts the interpretation of the findings. Including a comparison group would enhance the identification of deaths related to suicide, DoD, and natural causes within both groups. Moreover, an analysis of the risk factors associated with natural deaths in individuals with a history of SUD, excluding DoD-related deaths, should be conducted.

In lines 139 and 140, the manuscript should specify the variables studied at baseline rather than merely listing exceptions.

About the outcome variables, Table 1 presents more variables than those mentioned in section 2.3.2. This inconsistency between the data shown in Table 1 of the results section and the materials and methods section must be addressed.

Table 1 includes subscripts "a," "e," and "f," but there are no corresponding footnotes to explain these subscripts.

Additionally, the results do not report data on other relevant variables such as risk factors, methods of suicide, previous suicide attempts, education level, and age at the time of the suicide attempt.

The limitations of the DoD concept and its associated risk factors are particularly relevant as it applies to middle-aged white men.

The conclusions drawn in the manuscript are well-supported by the obtained data.

Furthermore, references 15 and 22 cite the same book, which should be corrected.

Author Response

Response to Reviewer

We sincerely appreciate the constructive feedback provided by the reviewer. Below, we'd like to address each comment point by point and indicate where changes have been made in the manuscript.

Comment 1: The DoD concept is limited as it was primarily developed to explain excess mortality among middle-aged white men. It does not fully capture early deaths linked to psychiatric conditions, such as homicide, cardiovascular disease, and metabolic disorders.

Response 1: We acknowledge this limitation and have added a discussion in the introduction highlighting that DoD may not fully account for the complexity of premature mortality among individuals with psychiatric disorders.

Manuscript change:
Expanded the introduction to address the limitations of the DoD concept and its relevance in this study population lines 78-86.

Comment 2: The methodology is clear, but the absence of a control group of individuals treated in the emergency department without a suicide attempt limits interpretation. A comparison group would allow for identifying differences in causes of death between groups. Moreover, an analysis of the risk factors associated with natural deaths in individuals with a history of SUD, excluding DoD-related deaths, should be conducted.

Response 2: We agree that a control group would have strengthened the findings. However, this study is based on an existing cohort, and comparisons with individuals treated in the emergency department without a suicide attempt were not possible within the available dataset. We now clarify this more explicitly in the limitations section. Additionally, while we acknowledge the importance of understanding risk factors associated with natural deaths in individuals with a history of SUD, excluding DoD-related deaths, such an analysis does not align with the primary aim of this study. The concept of Deaths of Despair (DoD) specifically focuses on deaths related to suicide, drug overdose, and alcohol-related conditions. Analyzing natural deaths separately would shift the study focus away from DoD, which aims to capture excess mortality driven by socioeconomic distress and behavioral health crises. Future research should explore the broader context of natural deaths among individuals with a history of SUD, but such an investigation would require a different analytical framework beyond the scope of this article.

Manuscript change: Added discussion in the limitations section about the absence of a control group and its impact on the interpretation of findings, lines 366-375. Also clarified why an analysis of risk factors for natural deaths is beyond the scope of this study while recognizing its importance for future research, lines 414-418.

Comment 3: In lines 139–140, please specify which variables were assessed at baseline instead of only listing exceptions.

Response 3: We have now provided a more detailed description of the baseline variables measured in the study to improve clarity.

Manuscript change: Updated the methods section to specify all baseline variables, lines 148-149.

Comment 4: Some variables in Table 1 are not described in the methods section.

Response 4: Thank you for pointing this out. We have now ensured that all variables presented in Table 1 are also described in the methods section. Manuscript change: Expanded the methods section to include all variables from Table 1.

Comment 5: Table 1 contains subscripts (a, e, f) that are not explained.

Response 5: Thank you for bringing this to our attention, these were not meant to be included and have now been removed.

Comment 6: The results do not report data on other relevant variables such as risk factors, methods of suicide attempt, previous suicide attempts, education level, and age at attempt.

Response 6: Thank you for this comment. First of all, age at attempt is included in the Cox regression models through the "age as time scale" method which we had accidentally failed to mention in the text. This method means that we fully adjust for any non-linear relationship between age at attempt and the outcome without it costing any degrees of freedom, but we don't get a numerical estimate of the effect of age at attempt. Beacuse this wasn't of particular interest for the research question, we considered this to be a good trade-off. We have now described this methodological choice in the statstics section.

Regarding the suggestion to include other relevant risk factors in the regression models, we have chosen not to do that for a particular reason. Because the number of events is small, we have to put a limit on the number of variables that can be included in the regression models. We therefore chose a limited set of variables to include, and because we wanted the four Cox regression models to be comparable, we chose to include the same variables in all the analyses. We acknowledge that this is a limitation of the study, and it has been included in the limitations section.

As for descriptive data of the suggested variables for the study participants, we refer to our previous papers based on the same data.

Because of the limited number of cases in the non-suicide death of despair analysis (n = 35), we had to severly limit the number of independent variables in our analyses.

Manuscript change: See method section, lines 205-213 and limitations lines 376-382.

Comment 7: Reference 15 and 22 appear to be duplicates.

Response 7: Thank you for catching this error. We have corrected the reference list and removed the duplicate reference. Manuscript change: Updated the reference list to eliminate duplication.

Round 2

Reviewer 3 Report

Comments and Suggestions for Authors

The authors have addressed the comments made in the initial review. Most of these limitations were addressed in the corresponding section of the manuscript.

I recommend a more in-depth discussion of the main findings. On the one hand, there is a lack of association of SUD as a risk factor for suicide. On the other hand, there is a lack of identification of relevant risk factors for SUD.

I recommend carefully reviewing the text, as punctuation marks are omitted in some sentences.

Author Response

We greatly appreciate the reviewer’s valuable comments, which have substantially helped us enhance the manuscript. Below, we provide a detailed point-by-point response addressing each comment and specifying revisions with corresponding page and line references to facilitate review:

Reviewer Comment 1:
"I recommend a more in-depth discussion of the main findings. On the one hand, there is a lack of association of SUD as a risk factor for suicide. On the other hand, there is a lack of identification of relevant risk factors for SUD."

Our Response:
We have expanded our discussion extensively to address these points in multiple sections:

  • We elaborated on the lack of association between substance use disorders (SUD) and suicide risk by discussing several possible explanations. These include the homogeneity of our sample consisting exclusively of individuals hospitalized after a suicide attempt (potentially limiting the impact of substance use as an isolated factor), potential attenuation of the association due to changes in substance use and treatment access over the long follow-up period, and methodological issues such as underdiagnosis at the time of admission (see page 8, lines 283–291).
  • We expanded the discussion about the concept of "deaths of despair" (DoD), explicitly addressing how our results both support and challenge aspects of this theoretical framework. We highlighted that the overall risk for DoD was similar among individuals with and without substance use disorders, supporting the concept's emphasis on common underlying psychosocial distress. However, we also clarified that our findings reveal important distinctions between suicide and non-suicidal DoD outcomes, indicating distinct psychiatric and psychosocial factors rather than a unified phenomenon (see page 9, lines 324–336)
  • We specifically discussed methodological considerations regarding the classification of deaths, emphasizing that we did not include deaths classified with undetermined intent (ICD-10 codes Y10–Y34) as suicide, which may contribute to an underestimation of suicide rates. We expanded the discussion on the challenges of accurately distinguishing between intentional suicide and accidental overdoses (see page 9, lines 353–359)
  • Regarding risk factors for SUD, we clarified our intention behind this analysis. Although our study was not primarily designed to identify specific SUD risk factors, we emphasized the importance of incorporating detailed substance use patterns (e.g., severity, type of substance) and social determinants (e.g., education, marital status, housing, income, social networks) in future research. This would enhance our understanding of potential confounders and the complex interplay influencing mortality risks (see page 11, lines 439–445).

Reviewer Comment 2:
"I recommend carefully reviewing the text, as punctuation marks are omitted in some sentences."

Our Response:
We have carefully reviewed the manuscript in its entirety, correcting punctuation and grammatical errors to enhance readability and clarity. These corrections have been made throughout the manuscript (pages 1–13).

We believe these detailed revisions have significantly improved the manuscript and thoroughly address the reviewer’s thoughtful comments. Thank you again for your valuable feedback.